# Role of Waste Cost in Thermoeconomic Analysis

**DOI:** 10.3390/e22030289

**Published:** 2020-03-02

**Authors:** Cuneyt Uysal, Ho-Young Kwak

**Affiliations:** 1Automotive Technologies Program, TOBB Vocational School of Technical Sciences, Karabuk University, 78050 Karabuk, Turkey; cuneytuysal@karabuk.edu.tr; 2Mechanical Engineering Department, Chung-Ang University, 84 Huksuk-ro, Dongjak-gu, Seoul 06974, Korea; 3Blue Economy Strategy Institute Co. Ltd. 23-13 Hyoryong-ro, 60-gil, Seocho-gu, Seoul 06721, Korea

**Keywords:** allocation of the waste cost, thermoeconomic theory, SPECO method, MOPSA method, unit cost of product

## Abstract

Power plants or thermal systems wherein products such as electricity and steam are generated affect the natural environment, as well as human society, through the discharging of wastes. The wastes from such plants may include ashes, flue gases, and hot water streams. The waste cost is of primary importance in plant operation and industrial ecology. Therefore, an appropriate approach for including waste cost in a thermoeconomic analysis is essential. In this study, a method to take waste cost into account in thermoeconomics to determine the production cost of products via thermoeconomic analysis is proposed. The calculation of the waste cost flow rates at the dissipative units and their allocation to system components are important to obtain the production cost of a plant.

## 1. Introduction

Thermoeconomic analysis is based on combining exergy analysis and economic principle. It provides a technique to evaluate the cost of inefficiencies or of individual process streams, including those of intermediate and final products [1]. The general objectives of a thermoeconomic analysis are (i) to reveal the cost formation process and (ii) to calculate the cost per exergy unit of the product streams of the system [2]. In the literature, several methods are reported for thermoeconomic analysis, such as specific exergy costing (SPECO) [2], engineering functional analysis (EFA) [3], last-in-first-out principle (LIFO) [4], theory of exergetic cost (TEC) [5], thermoeconomic functional analysis (TFA) [6], modified productive structure analysis (MOPSA) [7], and exergy-cost-energy-mass analysis (EXCEM) [8].

However, different thermoeconomic methods yield slightly different unit costs of products, mainly owing to the diverse methods used for the calculation of waste costs and their allocation to system components. Uysal et al. [9] conducted a thermoeconomic analysis of a coal-fired power plant by SPECO and MOPSA. The unit cost of the electricity generated by the system was obtained as 12.14 $/GJ with the SPECO method, whereas it was 14.06 $/GJ by the MOPSA method, with the unit specific exergy cost of coal of 4.26 $/GJ. However, the SPECO and MOPSA methods yielded similar unit costs of steam but significantly different unit costs of electricity from a CGAM plant, a predefined cogeneration system [10]. For a biogas engine powered cogeneration, it was also confirmed that the cost of the product varies with the thermoeconomic methodology [11].

The unit costs of products from a cogeneration plant may depend on the thermodynamic models and the cost allocation methods. Santos et al. [12] investigated the influence of various thermodynamics models on the cost allocation of a gas turbine cogeneration plant, using the internal energy-flow work-entropy (UFS) model [13]. They concluded that the higher the unit cost of electricity, the lower the unit cost of heat, and vice versa. Silva et al. [14] compared five allocation techniques generally employed in life-cycle assessment with three thermoeconomic allocation techniques used for multiproduct systems, employing the UFS model. It was reported that the thermoeconomic methods provided relatively less variation and yielded a more rational approach because the multiproduct step was disaggregated into its subsystems. Fortes et al. [15] employed the UFS thermoeconomic methodology for allocating costs and environmental impacts in a heat pump used in food dehydration and water production. It was found that the food dehydration and water production costs were 51% and 49% of the system cost, respectively. However, our main focus is the calculation of the unit costs of products from a coal-fired power plant, and a cogeneration plant, CGAM, which generates a single product or multiple products, using existing thermoeconomic methods with the same thermodynamic model.

The cost rate of irreversibility (lost work) related to the entropy generation occurring in components is one of the main parameters affecting the unit costs of products. In the MOPSA method, the cost flow rate of irreversibility, which is included in the cost-balance equation, acts as an input cost (source term), similar to the levelized cost of equipment. Concurrently, it is not treated in the cost-balance equation in the SPECO method. However, the SPECO method provides waste costs eliminated at dissipative units. The differences between these two methods in the costing of irreversibility were studied in detail by Uysal et al. [16].

Because the SPECO method provides the cost flow rate for any material stream at any state point in a plant, the costs of wastes that are eliminated at dissipative units, such as condenser and chimney, can be calculated by this method [9]. The allocation of the waste costs to system components is not implemented in the SPECO method. However, the summation of all the cost flow rates related to the irreversibility in system components is eliminated at the “boundary”, which acts as a dissipative unit in the MOPSA method. Thus, the total cost flow rate of irreversibility in plant can be considered as waste cost, and the cost flow rate of irreversibility for each component can be considered as the allocated waste cost to the system components in proportion to its lost work in the MOPSA.

Torres et al. [17] emphasized the importance of the cost formation of waste and proposed a mathematical structure required to calculate the waste cost. Agudelo et al. [18] proposed a methodology for allocating waste cost in thermoeconomic analysis. Allocation of the waste cost to the contributing productive components in proportion to their exergy destruction was suggested by Dobrovicescu et al. [19]. Seyyedi et al. [20] suggested a calculation method for determining the waste-cost distribution ratios based on the entropy distribution ratios in a system. However, these authors did not consider a relation between the waste costs eliminated at the dissipative units and their allocation to the system components. Even though these authors considered the distribution of the waste cost to system component, they did not discuss the elimination of the waste cost at a dissipative unit.

In this study, the costs of the wastes eliminated from dissipative units and their allocation to system components were considered for a coal-fired power plant and a gas turbine cogeneration plant, the CGAM system using the SPECO and MOPSA thermoeconomic methods. It was found that the calculated unit costs of products by SPECO and MOPSA were the same when the allocation of the waste costs eliminated at the dissipative units to the system components was conducted in the SPECO method. 

## 2. Thermoeconomic Methodologies

### 2.1. General Cost-Balance Equation for an Equipment in Power Plant

Even though the proposed thermoeconomic methodologies have different assumptions and applications methods, the cost-balance equation for each component should have the form given below. This is because the production cost of a component should include the cost of resources, as well as the cost of wastes [17].
(1)$${\dot{C}}_{k}^{P}={\dot{C}}_{k}^{F}+{\dot{C}}_{k}^{R}+{\dot{Z}}_{k}$$

Here, the superscripts *P*, *F*, and *R* denote the product, fuel or source, and wastes (residues), respectively; ${\dot{C}}_{k}^{R}$, which can be regarded as “source term” in the cost-balance equation, represents the cost of wastes charged for the k-th component; and ${\dot{Z}}_{k}$ represents the flow rate of the levelized cost, which includes all the financial charges associated with the owning and operating of the *k*-th plant component. The allocation of the cost flow rate of wastes can be achieved by several approaches [18,19,20].

Concurrently, the summation of the cost-balance equations for all the components, or the overall cost-balance equation should have the following form if the plant has one product, e.g., electricity [21,22]:(2)$${\dot{C}}^{W}={\dot{C}}^{CHE}+{\sum \dot{Z}}_{k}$$
where ${\dot{C}}^{W}$ and ${\dot{C}}^{CHE}$ represent the cost flow rates of work and fuel, respectively. Equation (2) can be rewritten explicitly as follows:(3)$$C_{W}{\dot{E}}_{net}^{W}=C_{o}{\dot{E}}_{c}^{CHE}+\sum {\dot{Z}}_{k}$$
where ${\dot{E}}_{net}^{W}$ and $C_{W}$ represent the rate of the work production and the corresponding unit cost. Terms ${\dot{E}}_{x}^{CHE}$ and $C_{o}$ represent the rate of the fuel provided and the corresponding unit cost. The overall cost-balance equation does not include the unit costs of mechanical exergy (*C_P_*), thermal exergy (*C_T_*), and lost work (*C_S_*), because these unit costs vanish during the addition process. These unit exergy costs may be considered as internal parameters in determining the production cost in MOPSA [23]. The summation of allocated waste costs in the system components, $\sum {\dot{C}}_{k}^{R}$ and the waste costs eliminated at the dissipative units should be null to obtain Equation (2). In the following, we discuss how two different thermoeconomic analysis methods, SPECO and MOPSA, deal with the waste cost in a coal-fired power plant and the CGAM system.

### 2.2. Specific Exergy Costing (SPECO)

The SPECO method is based on the modeling of the fuel and product definitions of a system by considering the exergy additions and removals from the system. In the SPECO method, the unit exergy cost value of each state located in the system is calculated by applying the basic principles from business administration. For a system that gains heat from the environment and produces work, the general cost-balance equation of the SPECO method can be written as follows [2]: (4)$$\sum {\left({\dot{C}}_{in}\right)}_{k}+{\dot{C}}_{q}+{\dot{Z}}_{k}=\sum {\left({\dot{C}}_{out}\right)}_{k}+{\dot{C}}_{W}$$
where *W* is the work, *q* is the heat, and subscripts *in* and *out* denote the inlet and outlet flow streams, respectively.

By applying the general cost-balance equation to each piece of equipment, an equation is obtained, so that the number of equations equal to the number of equipment can be obtained. However, in the SPECO method, the number of equations should be equal to the number of states in the system. Hence, auxiliary equations are required. The auxiliary equations can be obtained by considering the fuel (F) and product (P) rules. According to the F-rule, the total cost regarding the removed exergy is equal to the cost at which the extracted exergy is provided to the same flow in the next equipment. The P-rule proposes that exergy terms existing as a summation in the product definition of a component have the same average cost. By employing the F- and P-rules, the same number of equations with the number of states located in the system can be obtained.

### 2.3. Modified Productive Structure Analysis (MOPSA)

The MOPSA method was first introduced by Kim et al. [7]. In this method, the thermomechanical exergy is divided into its thermal and mechanical components. Including the exergy losses due to heat transfer through the nonadiabatic component, a general exergy-balance equation may be written as [24]
(5)$$\begin{array}{l} {\dot{E}}_{x}^{CHE}+{\dot{E}}_{x}^{BQ}+(\sum_{inlet}{\dot{E}}_{x,i}^{T}-\sum_{outlet}{\dot{E}}_{x,j}^{T})+(\sum_{inlet}{\dot{E}}_{x,i}^{P}-\sum_{outlet}{\dot{E}}_{x,j}^{P})\\ \;+T_{o}(\sum {\dot{S}}_{i}-\sum {\dot{S}}_{j}+{\dot{Q}}_{cv}/T_{o})={\dot{E}}_{x}^{W} \end{array}$$

The term including the entropy flow in Equation (5) represents the rate of irreversibility with minus sign. In the exergy-balance equation, the irreversibility due to the entropy generation acts as product.

In the MOPSA method, a unit exergy cost is assigned to each exergy components. In addition, the unit exergy cost is also adopted for the exergy destruction term. By assigning the unit exergy costs to the decomposed exergy in the stream, the cost-balance equation corresponding to the exergy-balance equation may be written as follows [24]:(6)$$\begin{array}{l} C_{o}{\dot{E}}_{x}^{CHE}+C_{T}(\sum_{inlet}{\dot{E}}_{x,i}^{T}-\sum_{outlet}{\dot{E}}_{x,i}^{T})+C_{P}(\sum_{inlet}{\dot{E}}_{x,i}^{P}-\sum_{outlet}{\dot{E}}_{x,i}^{P})\\ +C_{S}\cdot T_{0}(\sum_{inlet}{\dot{S}}_{i}-\sum_{outlet}{\dot{S}}_{j}+{\dot{Q}}_{cv}/T_{o})+{\dot{Z}}_{k}=C_{W}{\dot{E}}_{x}^{W}, \end{array}$$
where *C_o_*, *C_T_*, *C_P_*, and *C_S_* are the unit costs assigned to the chemical, thermal, and mechanical exergy flows and the negative value of irreversibility, respectively. In cost-balance equation, the cost of irreversibility acts as source in MOPSA. Equations (5) and (6) are the two basic equations used for determining the exergy cost of the products. The exergy-costing procedure by MOPSA is similar to that suggested by Lozano and Valero [25]. In fact, the productive structure of the system can be obtained by applying Equation (5) to each component. As can be seen in Equation (6), a single unit cost is assigned to the specific exergy, regardless of the type of exergy stream or the state of the stream. Applying the cost-balance equation given in Equation (6) to each component, Equation (6) can be rewritten as follows [26]:(7)$${\dot{C}}_{k}^{P}={\dot{C}}_{k}^{F}+{\dot{C}}_{S,k}+{\dot{Z}}_{k}$$
where superscripts F and P denote the fuel and the product for the *k*-th component, respectively. The third term in Equation (7) represents the lost cost flow rate corresponding to the entropy generation occurring in the component. Comparing Equation (1) with Equation (7), the term ${\dot{C}}_{S,k}$ in Equation (7) is the cost of the wastes or residues allocated to *k*-th component in proportion to their entropy generation [24]. 

By applying this general cost-balance equation of each system equipment, an equation set including the same number of equations with the number of system equipment is obtained. To solve this equation set, the number of equations should be equal to the number of unknown parameters. For this purpose, the auxiliary equations are required, and the junctions are used to obtain the auxiliary equations. The junctions are fictitious equipment where homogenous productions of two or more components merge [25]. In addition, the MOPSA method allows for the writing of one more cost-balance equation for the system boundary, which acts as a dissipative unit [24]. In the cost-balance equation written for the system boundary, only gas and fluid streams crossing the system boundary are considered, and it is assumed that the principal product of system boundary is the entropy generation.

## 3. Case Studies

### 3.1. Plants Description

A schematic of the coal-fired power plant that was studied thoroughly by Uysal et al. [9], using SPECO and MOPSA, is displayed in Figure 1. The system includes 26 pieces of equipment. Combustion air was supplied via primary (FAN-1) and secondary air fans (FAN-2). FAN-1 and 2 supplied air with mass flow rates of 62.27 and 100.83 kg/s, respectively. Before entering a coal-fired steam boiler (SB), the air was preheated in an air preheater (AP). Coal was supplied to the SB, with a mass flow rate of 14.78 kg/s. The flue gas emitted at the SB was used to superheat the steam generated by it and to preheat the combustion air. The steam generated by the SB was transferred to a cyclone (CYC) and then to superheaters (SH1, SH2, and SH3), where it was superheated via the flue gas in these components. The superheated steam leaving SH3 entered a high-pressure turbine (HPT). A part of the steam leaving the HPT was transferred to a heat exchanger (RH), where it was reheated. The remaining part of the steam leaving the HPT was transferred to a high-pressure heat exchanger (HPH2) for preheating the water supplied to the SB. The reheated steam in the RH was transferred to a low-pressure turbine (LPT). The shaftworks generated by the LPT and HPT produced electricity in a generator (G). A part of the electricity generated by the system was supplied to circulation pumps (PUMP1, PUMP2), a condenser pump (CP), and fans (FAN1, FAN2) to drive them. The remaining part of the electricity generated by the system was supplied to the electric network. A part of the steam leaving the LPT entered a condenser (COND), and the remaining part was used to preheat the water transferred to the SB. COND was water-cooled, and an environmental water source was used as the heat sink. The water was sent to a condenser tank (CT), where it was mixed with the steam exiting a low-pressure heat-exchanger unit (LPH1, LPH2). Accordingly, the steam coming from LPH1 and LPH2 was condensed. The water was pumped through LPH1 and LPH2 to a feedwater tank (FWT) via the CP. The FWT collected some water or steam streams used as hot streams in the RHs. The water storage in the FWT was pressurized via PUMP1 and PUMP2. The pressurized water entered HPH1 and HPH2, and it was preheated through HPH1, HPH2, and a deaerator (DPH), respectively. After this process, the preheated water stream was transferred to economizer units (ECO1, ECO2), where it was reheated. The reheated water stream leaving ECO2 was supplied to the SB. Accordingly, the cycle was completed. The flue gas emitted by the SB was used as hot stream in CYC, SH3, RH, SH2, SH1, ECO2, and ECO1, respectively. Finally, it was released to the environment through the AP. 

The CGAM system refers to a cogeneration plant that delivers electricity of 30 MW and saturated steam of 14 kg/sat 20 bar. A schematic of the cogeneration plant is depicted in Figure 2. The system consists of an air compressor (AC), an air preheater (APH), a combustion chamber (CC), a gas turbine (GT), and a heat-recovery steam generator (HRSG). The environmental conditions are defined as *T_o_* = 298.15 K and *P_o_* = 1.013 bar. The parameters of the CGAM system used to facilitate the comparisons of the exergy-costing methodologies were presented by Valero et al. [27]. 

### 3.2. Overall Cost-Balance Equation of Coal-Fired Power Plant and CGAM System by SPECO and MOPSA

For the coal-fired power plant displayed in Figure 1, 27 equations with 62 unknown cost flow rates from the 26 components and a turbine group can be obtained by SPECO. To solve the unknown 62 cost flow rates, 35 auxiliary equations are required. The summation of all the cost-balance equations with the help of the auxiliary equations obtained for the coal-fired power plant [9] yields the following equation:(8)$$\begin{array}{c} C_{W}{\dot{E}}_{net}^{W}(={\dot{C}}_{56}-{\dot{C}}_{58}-{\dot{C}}_{59}-{\dot{C}}_{60}-{\dot{C}}_{61}-{\dot{C}}_{62})=C_{o}{\dot{E}}_{x}^{CHE}(={\dot{C}}_{7})\\ +\sum {\dot{Z}}_{k}-({\dot{C}}_{16}+{\dot{C}}_{42}) \end{array}$$

In Equation (8), the first and second terms represent the cost flow rates of the electricity and fuel, respectively, and the third term represents the cost flow rate of the levelized cost of all the equipment. It can be seen from Equation (8) that the material streams at the AP (16 in Figure 1) and COND (42 in Figure 1) are waste streams and components AP and COND act as dissipative units. Moreover, the summation of all the cost-balance equations, with the help of the auxiliary equations obtained by MOPSA for the coal-fired power plant, which has a single product, yields Equation (3).

For the CGAM system, the summation of all the cost-balance equations obtained by the SPECO method yields the following equation:(9)$$C_{W}{\dot{E}}_{net}^{W}+C_{BQ}{\dot{E}}_{x}^{BQ}=C_{o}{\dot{E}}_{x,12}^{CHE}+C_{M}{\dot{E}}_{x,12}^{M}+\sum {\dot{Z}}_{k}-{\dot{C}}_{7}$$

The first two terms represent the cost flow rate of electricity and steam, respectively; the next two terms indicate the cost flow rates of chemical and mechanical exergies of fuel, and the fifth term indicates the levelized cost flow rate of equipment. The last term represents the cost flow rate of flue gases leaving the dissipative unit of HRSG. Meanwhile, the summation of all the cost-balance equations obtained by the MOPSA is given by the following:(10)$$C_{W}{\dot{E}}_{net}^{W}+C_{BQ}{\dot{E}}_{x}^{BQ}=C_{o}{\dot{E}}_{x,12}^{CHE}+C_{M}{\dot{E}}_{x,12}^{M}+\sum {\dot{Z}}_{k}$$

For the CGAM system without the HRSG, the summation of all the cost-balance equations obtained by the SPECO can be written as follows.
(11)$$C_{W}{\dot{E}}_{net}^{W}=C_{o}{\dot{E}}_{x,12}^{CHE}+C_{M}{\dot{E}}_{x,12}^{M}+\sum {\dot{Z}}_{k}-{\dot{Z}}_{HRSG}-{\dot{C}}_{6}$$

The last term represents the cost flow rate of flue gases at AP, which acts as the dissipative unit for the gas turbine plant. Meanwhile, the summation of all the cost-balance equation obtained by the MOPSA is as follows:(12)$$C_{W}{\dot{E}}_{net}^{W}=C_{o}{\dot{E}}_{x,12}^{CHE}+C_{M}{\dot{E}}_{x,12}^{M}+\sum {\dot{Z}}_{k}-{\dot{Z}}_{HRSG}$$

A comparison between the SPECO and MOPSA of the overall cost-balance equation for the coal-fired power plant, CGAM system, and gas turbine plant provides a clear difference between two thermoeconomic methods. One can obtain the unit cost of product by the MOPSA with known value of cost flow rates of fuel and the levelized cost flow rate of plant. However, the SPECO needs to know additional information on the cost flow rate leaving the dissipate units to obtain the unit cost of products.

Equation (3), obtained by MOPSA for the coal-fired power plant, indicates that the summation of all the cost flow rates for the thermal, mechanical exergies and irreversibility is null. It can be written explicitly, as follows:(13)$$\sum {\dot{C}}_{k}^{F}=C_{o}{\dot{E}}_{x}^{CHE}$$
(14)$$\sum {\dot{C}}_{k}^{P}=C_{W}{\dot{E}}_{x}^{W}$$
(15)$$\sum {\dot{C}}_{S,k}+{\dot{C}}_{S,boundary}=0$$

Using Equation (15), Equation (3) can be rewritten as follows: (16)$$C_{W}{\dot{E}}_{net}^{W}=C_{o}{\dot{E}}_{x}^{CHE}+\sum {\dot{C}}_{S,k}+\sum {\dot{Z}}_{k}+{\dot{C}}_{S,boundary}$$

Equation (16) indicates that the cost flow rate related to the waste flow streams dissipated in the boundary, ${\dot{C}}_{S,boundary}$, should be allocated to each component as the source, $\sum {\dot{C}}_{S,k}$, which was achieved in the cost-balance equation in MOPSA. Even though SPECO is successful in obtaining the cost flow rate of wastes at dissipative units, it fails to distribute the waste cost to the appropriate component in the power plant.

## 4. Calculation Results

### 4.1. Unit Costs of Products Calculated by the SPECO and MOPSA

Coal was supplied to the SB with a mass flow rate of 14.78 kg/s in the coal-fired power plant. The lower heating value (LHV) of coal used in the plant was approximately 23228.3 kJ/kg. The unit cost of coal is 4.26 $/GJ. The electricity produced by the coal-fired power plant was approximately 143,250 kW. An amount of 7220 kW electricity was used as feedback to operate the pumps and fans. The remaining portion was supplied to the electrical network. The estimated levelized cost flow rate of all the components of the plant was approximately 1323.16 $/h. As discussed in the previous section, the SPECO method is successful at obtaining the cost flow rate of the waste of flue gases at 16 and hot water stream at 42. The cost flow rate of flue gases eliminated at AP and COND in the coal-fired power plant were approximately 224.13 and 705.35 $/h, respectively. With these waste cost flow rates at the dissipative units, the unit cost of electricity of the coal-fired plant from Equation (8) is approximately 12.14 $/GJ, which is lower than that of electricity from Equation (3), 14.06 $/GJ, by MOPSA.

For the CGAM system having net power of 300 MW, the calculated cost flow rates of mechanical and chemical exergy of the fuel are 63.18 and 1209.32 $/h, respectively, and the estimated levelized cost flow rate of all the components of the CGAM is approximately 138.74 $/h [28]. The calculated cost flow rates of the steam and the waste of flue gases leaving the HRSG are 479.45 and 88.93 $/h, using the SPECO method, respectively [28]. Using these values, the unit cost of electricity, calculated by Equation (9), is approximately 7.80 $/GJ. This unit cost of electricity by SPECO is lower than that calculated by Equation (10), using MOPSA, 8.48 $/GJ [10].

The CGAM system without the HRSG becomes a gas-turbine power plant having net power of 300 MW. The levelized cost flow rate of HRSG is 16.93 $/h [28]. The cost flow rate of the waste eliminated at the AP is approximately 551.44 $/h by SPECO [9]. The unit cost of electricity, calculated by Equation (11) is approximately 7.80 $/GJ, which is significantly lower than that by MOPSA, 12.91 $/GJ [10].

For all the cases considered in this study, the unit cost of products calculated by SPECO turned out to be less than the ones by MOPSA. This is because an appropriate allocation of the waste cost flow rate eliminate at the dissipative unit is not conducted in SPECO.

### 4.2. A Remedy for the Thermoeconomic Method

Equations (8), (9) and (11), which are obtained by the summation of all the cost-balance equations, using the SPECO method, indicate that the waste cost leaving the dissipative units should be allocated to the plant components: (17)$$\sum {\dot{C}}_{k}^{R}-({\dot{C}}_{16}+{\dot{C}}_{42})=0\;\mathrm{for}\;\mathrm{coal}-\mathrm{fired}\;\mathrm{plant}$$
(18)$$\sum {\dot{C}}_{k}^{R}-{\dot{C}}_{7}=0\;\mathrm{for}\;\mathrm{CGAM}\;\mathrm{system}$$
(19)$$\sum {\dot{C}}_{k}^{R}-{\dot{C}}_{6}=0\;\mathrm{for}\;\mathrm{the}\;\mathrm{CGAM}\;\mathrm{system}\;\mathrm{without}\;\mathrm{the}\;\mathrm{HRSG}$$

With Equations (17)–(19), the overall cost-balance equations given in Equations (8), (9) and (11) reduce to Equations (3), (10) and (12), respectively, which might be correct for a plant having a single product [21,22] and the CGAM system having multiple products. The waste cost flow rates leaving the AP and COND in the coal-fired power plant, as calculated by SPECO is approximately 929.48 $/h, which is smaller than the waste cost flow rate leaving the boundary, 1293.29 $/h, by MOPSA. However, from the allocation of the waste cost flow rate to the components under the condition given in Equation (17), one may obtain the same unit cost of electricity as the one evaluated by MOPSA. For the CGAM system, the cost flow rate of the waste leaving the HRSG is approximately 88.93 $/h, as calculated by SPECO. The unit cost of electricity by the allocation of the waste to the system components under the condition given in Equation (18) is approximately 8.63 $/GJ, which is close to the unit cost of electricity by MOPSA, 8.48 $/GJ. The calculated unit cost of electricity by SPECO under the condition given in Equation (19) is approximately 12.9 $/GJ, which is the same value as that obtained by MOPSA, even though the calculated cost flow rates of the waste eliminated at the AP by SPECO (541.44 $/h) and MOPSA (781.76 $/h) are quite different from each other.

For reference purposes, the cost flow rates of the wastes leaving the AP and COND in a coal-fired power plant are allocated proportionally to the irreversibility occurring in the components, as listed in Table 1. The allocated waste cost flow rate for a component may be considered as its lost cost flow rate [16], and the waste cost flow rate eliminated at the dissipative units can be used as a measure of the effect on the environment. In the fifth column, a possible allocation of the waste cost flow rates eliminated by the AP and COND in proportion to the irreversibility occurring at each component for SPECO is displayed. 

Without elimination of the cost flow rate of the waste at the “boundary”, which acts a dissipative unit, the overall cost-balance equation for the power plant, given in Equation (3), becomes the following:(20)$$C_{W}{\dot{E}}_{net}^{W}=C_{o}{\dot{E}}_{x}^{CHE}+\sum {\dot{C}}_{S,k}+\sum {\dot{Z}}_{k}$$

Using Equation (20), the unit costs of products for the coal-fired power plant and CGAM are 15.94 and 13.9 $/GJ, respectively, which are higher than the cost of products estimated by Equations (3) and (10). These results indicate that the waste costs distributed in the system components should be eliminated at the dissipative unit boundary. In this case, the unit cost of product varies depending on the cost of waste.

## 5. Conclusions

The cost flow rate of waste and its allocation to the system components were considered for a
>coal-fired power plant and CGAM system, using SPECO and MOPSA thermoeconomic methodologies. To estimate the appropriate unit cost of product from power plant, the cost of waste eliminated at the dissipative units should be allocated to the system components. It is noted that the cost flow rate of allocated wastes to the system components acts as a source in the cost-balance equation. On the other hand, the cost flow rate of wastes eliminated at dissipative unit acts as a sink in the cost-balance equation and is considered to have a negative environmental impact. With appropriate consideration of the waste cost in thermoeconomic analysis, the same unit cost of products from the coal-power plant and the CGAM system, using SPECO and MOPSA, was obtained.

## Figures and Tables

**Figure 1 entropy-22-00289-f001:**
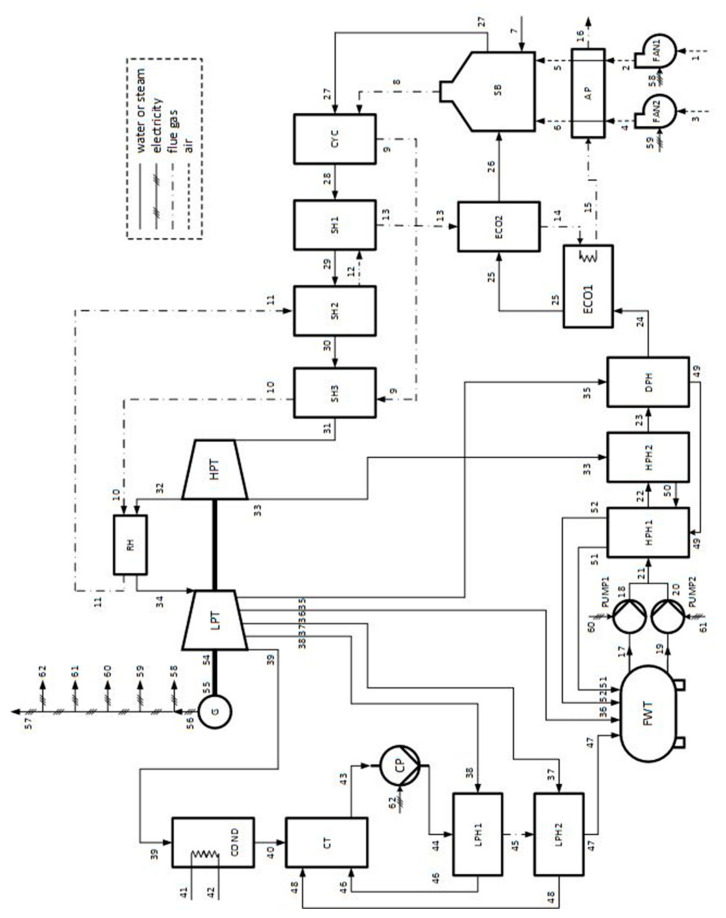
Schematic of the considered coal-fired power plant [9].

**Figure 2 entropy-22-00289-f002:**
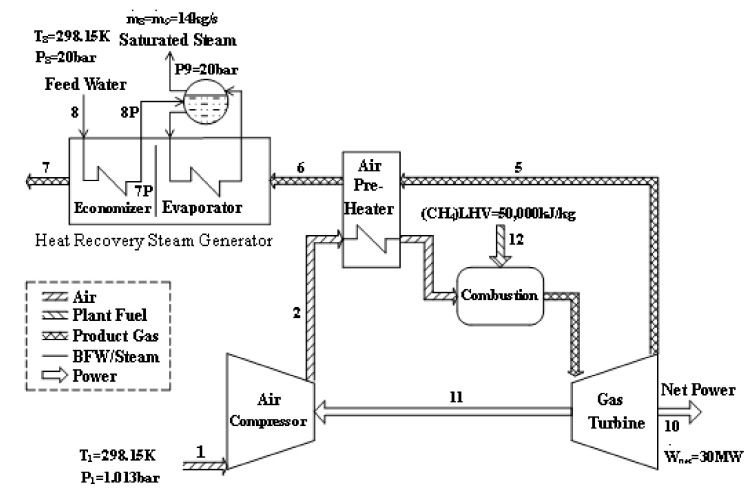
CGAM gas turbine cogeneration system [27].

**Table 1 entropy-22-00289-t001:** Cost flow rate of the work, fuel, steam, the allocated waste cost by SPECO, lost cost flow rate by MOPSA, and levelized cost in a coal-fired plant.

Equipment	${\dot{C}}^{W}\;(\$/h)$	${\dot{C}}_{o}\;(\$/h)$	${\dot{C}}_{BQ}\;(\text{\$}/h)$	${\dot{C}}_{k}^{R}\;(\mathrm{SPECO}:\;\$/h)$	${\dot{C}}_{S,k}\;(\mathrm{MOPSA}:\;\$/h)$	${\dot{Z}}_{k}\;(\$/h)$
FAN1	0	0	0	−2.14	−2.98	−3.6
FAN2	0	0	0	−1.39	−1.91	−3.6
AP	0	0	0	−9.11	−12.7	−7.21
SB	0	−5554.67	0	−710.59	−988.74	−83
CYC	0	0	0	−8.37	−11.58	−10.38
RH	0	0	0	−15.15	−21.14	−17.32
SH1	0	0	0	−6.41	−8.94	−7.21
SH2	0	0	0	−27.70	−38.59	−10.82
SH3	0	0	0	−28.81	−40.12	−7.21
ECO1	0	0	0	−6.51	−9.03	−6.85
ECO2	0	0	0	−3.16	−4.4	−6.85
PUMP1	0	0	0	−1.86	−2.64	−5.05
PUMP2	0	0	0	−1.95	−2.71	−5.05
HPH1	0	0	0	−4.09	−5.72	−6.49
HPH2	0	0	0	−21.94	−30.46	−6.49
DPH	0	0	0	−1.12	−1.6	−5.77
HPT	2236.4	0	0	−6.23	−8.72	−288.71
LPT	4806.73	0	0	−10.87	−15.16	−245.4
COND	0	0	851.17	−49.54	−68.94	−61.35
CT	0	0	0	−0.07	−0.09	−2.88
CP	0	0	0	−1.77	−2.47	−2.16
LPH1	0	0	0	−4.74	−6.54	−8.66
LPH2	0	0	0	−3.53	−4.89	−8.66
FWT	0	0	0	−1.30	−1.87	-3.6
G	−7043.13	0	0	−0.93	−1.27	−198.48
Boundary	0	0	−851.17	929.48	1293.29	−310.36

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
