# Peer review of "Role of Waste Cost in Thermoeconomic Analysis"

_entropy, 2020, doi:10.3390/e22030289_

Round 1
Reviewer 1 Report
This paper is an interesting work since it deals with the difficult and important problem of cost apportioning of the dissipative units and residues. The paper is relevant in the theme context.
English must be reviewed by a native proofreader. Some parts of the text and not understandable, for example: "It was found that the calculated unit cost of product by SPECO and MOPSA are the same when the allocation of waste costs eliminated at the dissipative units to the system components was done in SPECO method" and "Because SPECO method provided the cost flow rate for any material stream at any state point in plant, the cost of wastes which are eliminated at dissipative units such as condenser and chimney can be calculated by this method [8]. While the allocation of the waste cost to system components was not done in SPECO method." These make no sense! ENGLISH MUST BE REVIEWER THOROUGHLY!
This sentence is obvious: "However, different thermoeconomic methods produce somewhat 30 different unit cost of product, which is mainly due to the different methods in the calculation of waste costs and their allocation to the system components. "
Instead of lumping references such as "There are several methods for thermoeconomic analysis in the literature [2-7]. " Please refer to them individually. There is the theory of exergy cost, UFS, E&S...
Cite other work besides your own... for example:
CHAVES, A. F. F. ; SILVA, J. A. M. ; Carvalho, M. . Environmental impact and cost allocations for a dual product heat pump. ENERGY CONVERSION AND MANAGEMENT, v. 173, p. 763-772, 2018.
SILVA, J. A. M. ; SANTOS, J. J. C. S. ; CARVALHO, M. ; OLIVEIRA JUNIOR, S. . On the thermoeconomic and LCA methods for waste and fuel allocation in multiproduct systems. ENERGY, v. 127, p. 775-785, 2017.
AMORIM, LORENZONI RAPHAEL ; CONCEIÇÃO SOARES, SANTOS JOSÉ JOAQUIM ; BARBOSA, LOURENÇO ATILIO ; MARCON, DONATELLI JOÃO LUIZ . On the accuracy improvement of thermoeconomic diagnosis through exergy disaggregation and dissipative equipment isolation. ENERGY, v. 1, p. 116834, 2019.
SANTOS, R. G. ; FARIA, P. R. ; SANTOS, J. J. C. S. ; SILVA, J. A. M. ; DONATELLI, J. L. M. . THE EFFECT OF THE THERMODYNAMIC MODELS ON THE THERMOECONOMIC RESULTS FOR COST ALLOCATION IN A GAS TURBINE COGENERATION SYSTEM. Engenharia Térmica, v. 14, p. 47-52, 2015.
However, the paper was not well organized nor well structured. Generally, the thermodynamic and economic modeling is presented before the thermoeconomic approaches. This step is common for all thermoeconomic methodologies.
The basic principle of cost allocation is that "all cost entering the plant must be charged to the final products". Consequently, the cost of the residues and of the dissipative product must be internalized in the thermal system analyzed throughout some productive ones. In other words, in this case, different methodologies should provide different value of unit cost for the final products. However, the higher the cost of power, the lower will be the cost of heat, and vice-versa. The author did not present how the approaches deal with the dissipative components and residues.
It might be useful to the authors to read once again the papers they cited in order to better understand the application of these approaches: product and fuel definition, auxiliary equation formulation and dissipative components and residues treatment. These steps must be presented for both approaches. Not for SPECO only.
Finally, due to its relevance in this theme context of dissipative component and residues cost allocation, I strongly encourage the authors in order to carry out a careful analysis of these references related to the application of both thermoeconomic approaches in order to obtain corrects results and, consequently, improve the quality of the next version of the paper.
Discussion about the differences and similarities of the methods and their applicability range should be extended and improved.
The originality of the paper needs to be further clarified. It is of importance to have sufficient results to justify the novelty of a high quality journal paper.
An updated and complete literature review should be conducted to present the state-of-the-art and knowledge gaps of the research with strong relevance to the topic of the paper.
The results should be further elaborated to show how they could be used for real applications.
Reviewer 2 Report
The topic is convenient according to the scope of the journal and the presented methodology and results are fine. However, the major suggestion is the update of references. Even though the authors present a very clear summary of the fundamental background of exergoeconomics, it is known that the literature has also recent studies. For example (not only these, please update the references by checking the other studies as well):
1) https://doi.org/10.1016/S0196-8904(02)00179-6
2) https://doi.org/10.1016/j.energy.2008.07.018
3) https://doi.org/10.1016/j.enconman.2017.06.079
4) https://doi.org/10.1016/S0360-5442(02)00096-8
Also, as a minor suggestion, the resolution of the system schematic can be improved.
Reviewer 3 Report
In the paper “A Role of Waste Cost in Thermoeconomic Analysis”, the authors aimed at comparing/integrating two thermoeconomic methods, i.e. Specific Exergy Costing and Modified Productive Structure Analysis MPSOA, for “waste cost” allocation.
In my opinion, the paper is not good for publication and substantial modifications are needed before being resubmitted. The structure should be totally re-organized, in order to help reader in understanding what they investigated and the main results achieved. An accurate literature review concerning “waste cost” allocation criteria is due considering that this is an open issue in Thermoeconomics. Novelty of the proposed manuscript is not highlighted as well. It is not good to introduce the case study within the section focused on the description of case study. Lot of mistakeS are present. Results section is not evident.
Round 2
Reviewer 1 Report
Please check English language again, ask a native English proofreader to review the manuscript.
Author Response
An English editing specialist edited our manuscript.
Reviewer 3 Report
As already explained in my first revision, the description of the case study should be placed outside the method section. look at the structure of this paper https://doi.org/10.1016/j.energy.2012.07.034
Secondly, equations are shown along with results discussions, and this is not good in my opinion. Results sections must be well-identified.
Author Response
- In revised manuscript, the case study was placed outside the method section.
- Results section was added in revised manuscript.